# Detection of Hepatitis A Virus and Other Enteric Viruses in Shellfish Collected in the Gulf of Naples, Italy

**DOI:** 10.3390/ijerph16142588

**Published:** 2019-07-19

**Authors:** Giovanna Fusco, Aniello Anastasio, David H. Kingsley, Maria Grazia Amoroso, Tiziana Pepe, Pina M. Fratamico, Barbara Cioffi, Rachele Rossi, Giuseppina La Rosa, Federica Boccia

**Affiliations:** 1Department of Animal Health, Istituto Zooprofilattico Sperimentale del Mezzogiorno, Via Salute 2, Portici, 80055 Naples, Italy; 2Department of Veterinary Medicine and Animal Production, Università degli Studi di Napoli Federico II, Via Federico Delpino 1, 80137 Naples, Italy; 3U.S. Department of Agriculture, Agricultural Research Service, Delaware State University, Dover, DE 19901, USA; 4USDA, Agricultural Research Service, Eastern Regional Research Center, 600 E. Mermaid Lane, Wyndmoor, PA 19038, USA; 5Department of Environment and Health, Istituto Superiore di Sanità, Viale Regina Elena 299, 00161 Rome, Italy

**Keywords:** enteric viruses, mollusks, molecular methods, HAV

## Abstract

To assess the quality of shellfish harvest areas, bivalve mollusk samples from three coastal areas of the Campania region in Southwest Italy were evaluated for viruses over a three-year period (2015–2017). Screening of 289 samples from shellfish farms and other locations by qPCR and RT-qPCR identified hepatitis A virus (HAV; 8.9%), norovirus GI (NoVGI; 10.8%) and GII (NoVGII; 39.7%), rotavirus (RV; 9.0%), astrovirus (AsV; 20.8%), sapovirus (SaV; 18.8%), aichivirus-1 (AiV-1; 5.6%), and adenovirus (AdV, 5.6%). Hepatitis E virus (HEV) was never detected. Sequence analysis identified HAV as genotype IA and AdV as type 41. This study demonstrates the presence of different enteric viruses within bivalve mollusks, highlighting the limitations of the current EU classification system for shellfish growing waters.

## 1. Introduction

The presence of enteric viruses in water, food, and the environment is a global concern, prompting countries to protect public health through the development of strategic plans aimed at reducing epidemic cases in their territories. Hepatitis A virus (HAV) is considered one of the most serious food-borne viruses. HAV is transmitted primarily through contact with infected individuals or through the consumption of contaminated water and foods, such as raw or undercooked shellfish [1,2]. In Italy, human cases of hepatitis A have decreased over the past 30 years, and the incidence of HAV is now classified as medium to low (0.6–0.8 cases per 100,000 inhabitants) as reported by the Italian Surveillance System of Acute Hepatitis, SEIEVA (Integrated Epidemiological System for Acute Viral Hepatitis) in 2015 to 2016 (http://www.epicentro.iss.it) [3,4]. Although both genotype IA and IB are found throughout Italy, strain distribution is not homogeneous. In Southern Italy, most outbreaks of HAV have traditionally been from genotype IB strains, many of which are frequently associated with the consumption of raw seafood [5]. In Northern Italy, genotype IA is more predominant. A noteworthy genotype IA outbreak occurred in Northern and Central Italy during 2013 that was associated with the consumption of frozen berries [6]. Since 2016, an alarming increase in hepatitis A cases among male homosexuals has been observed in Italy, as well as in Europe overall. In Italy, the number of these cases for men aged 25 to 54 years is higher than in other European countries (SEIEVA 2017) [7]. During the period of July 2016 to July 2017 in northern Italy (Brescia), 42 cases were reported, of which, 25 (60%) were associated with sexual behavior (fecal-oral contact) rather than transmission of the pathogen through food [8]. This new trend is also confirmed by the data published by the European Center for Disease Prevention and Control (ECDC), which indicates that the risk factor related to sexual behavior is increasing with respect to the classic risk factors, such as the consumption of raw mollusks or travel to endemic areas [8]. The World Health Assembly has recently adopted a global hepatitis A strategy, with the goal of reducing HAV illness [7].

Among the other enteric viruses, norovirus (NoV) is an important agent responsible for large numbers of gastroenteritis cases. This virus is frequently transmitted through person-to-person contact, water, and foods, such as mollusks, which are frequently contaminated by exposure to urban wastewater. It is worth noting that an individual with an enteric virus infection eliminates between 10^5^ to 10^13^ viral particles per gram of stool [9], and thus, it follows that only a few infected people within a given population are sufficient to shed a huge charge of viral particles.

Other enteric viruses, such as aichivirus (AiV), sapovirus (SaV), astrovirus (AsV), adenovirus (AdV), and rotavirus (RV), can also potentially cause human illnesses [10]. Today, AiV is considered an emerging human enteric pathogen able to cause gastroenteritis through the consumption of contaminated shellfish and other foods [10]. Sapoviruses are commonly implicated in shellfish-associated outbreaks [11]. Adenovirus contamination is thought to be highly persistent in the environment and within shellfish, leading some researchers to propose it as a general index for virus contamination [12]. Although not frequently implicated in outbreaks, it has been implicated in shellfish-borne transmission [13]. Although shellfish-borne transmission of RV, AdV, and AsV is less common, it has occurred [10,12]. Identification of hepatitis E virus (HEV) within shellfish is uncommon and is not commonly associated with HEV transmission, but it is not unprecedented. The first report of HEV in European shellfish was by Béji-Hamza et al. [14] and La Rosa et al. [15], who confirmed the presence of HEV in shellfish from commercial harvesting areas and classified them as genotype G3. Enteric viruses eliminated in feces (e.g., HAV, NoV) can persist for long periods in water, wastewater, and in sewage sludge [16]. To control disease in humans, it is important to ensure that consumers have access to a safe water supply and that shellfish are reared in microbiologically-safe water. In this regard, it should be noted that current European legislation, Regulation CE 854/2004, Regulation CE 2285/2015, and Regulation CE 2073/2008 [17,18,19], only provides for the use of *E. coli* as an indicator organism to evaluate the degree of fecal contamination of water and the microbiological criteria applicable to food products, despite numerous authors indicating an absence of correlation between enteric virus and *E. coli* [1,10]. The objective of the current study was to investigate the presence of viral agents within the digestive tissues of mollusks produced and marketed in southern Italy. The present study focuses on the detection of HAV, NoV (GI and GII), AsV, RV, AiV, SaV, AdV, and HEV. 

## 2. Materials and Methods

### 2.1. Sampling

Over a three-year period from 2015 to 2017, a total of 289 1-kg samples of shellfish were collected, of which, 280 were mussels (*Mytilus galloprovincialis*) and 9 were clams (*Ruditapes philippinarum*). Samples were collected from various production sites on the southwestern coast of the Campania region of Southern Italy (Table 1 and Figure 1) by the veterinary services in charge of the territory under monitoring plans provided for by EC Regulation 854/2004 [17] and by EC Regulation 2285/2015 [18] for the classification of waters and microbiological food safety criteria applicable to food. Specifically, 146 samples (defined as 1 kg of shellfish) of bivalve mollusks were examined in 2015, while 77 and 66 samples were collected in 2016 and 2017, respectively. Overall, 46 samples (16.0%) were collected from class A sites, 237 (82.1%) from class B sites, 1 sample was from area C, and 5 from natural banks (site classification described in Reg. CE 854/2004 [17], as shown in Table 2). Essentially, under EU regulations, class A shellfish are considered safe enough to be directly sold and consumed, while class B shellfish require post-harvest treatment (depuration) prior to sale. Class C shellfish are generally not considered safe for human consumption unless extensively depurated. After collection, the mollusks were transported at 4 °C to the Virology Department of Experimental Zooprophylactic Institute of Southern Italy (IZSM), and then analysis was performed within 24 h.

### 2.2. Nucleic Acid Extraction and Detection of Viruses by Quantitative Real-Time PCR

Twenty mussels were randomly selected from each sample. Hepatopancreas samples were sectioned from individual animals, were finely chopped, and then pooled together. Two grams of the pool underwent viral recovery according to the ISO/TS 15216-2:2013 [20].

Nucleic acids were extracted from 400 µL of viral extract using a mengovirus clone (vMC_0_) as the nucleic acid extraction efficiency control. Extraction was carried out on the QIAsymphony SP automatic extraction system (Qiagen Inc., Hilden, Germany) with the DSP Virus/Pathogen Midi kit (Qiagen Inc., Hilden, Germany) following the manufacturer’s instructions. Nucleic acids were eluted in 60 µL of elution buffer and immediately analyzed by real-time PCR or stored at −80 °C until use. To determine the presence of the viral targets, quantitative one-step reverse transcription PCR (RT-qPCR) assays were performed on a 7500 Fast Real-Time PCR System (Applied Biosystems, Foster City, CA, USA) using the AgPath-ID One-step RT-PCR kit (Applied Biosystems by Thermo Fisher, Life Technologies Austin, TX, USA). Each RT-PCR assay for an individual viral target was performed in triplicate [2], with primers and probes specific for the virus investigated. HAV, NoVGI, and NoVGII were detected following the UNI CEN ISO/TS 15216-1:2013 method [20]. RV, AiV, HEV, SaV, AdV, and AsV were investigated according to protocols described in the literature [21,22,23,24,25,26]. The sequences of primers and probes are shown in Table 3.

For each virus, quantification analysis was done using a standard curve achieved from amplifying serial dilutions of the quantified standard plasmid DNA (from 1 to 1 × 10^6^ copies/reaction). Log genome copies were plotted against the C_t_ value, and results are expressed as the number of genome copies per gram of digestive tissue (copies/g) according to Fusco et al. [2].

The HEV positive control was a kind gift of the Federal Research Institute for Animal Health “FLI” (Germany). HAV, NoV GI, NoV GII, RV, and AdV quantified standard plasmids were kindly provided by the Italian National Reference Laboratory for the monitoring of viral contamination of bivalve mollusks (Istituto Superiore di Sanità, Rome). AsV and SV plasmid standards were purchased from CeeramTools (Biomerieux; Craponne, France). The AiV plasmid standard was kindly gifted by Dr. J.L. Romalde, Universidad de Santiago de Compostela, Spain. Results were analyzed as already described by Fusco et al. [2]. Briefly, when at least one replicate (out of the three performed) showed a C_t_ < 40, the sample was considered positive. Samples with a C_t_ ≥ 40 were considered negative.

### 2.3. Sequencing and Phylogenetic Analysis

Samples positive for HAV and adenovirus, exhibiting C_t_ values of ≤38, were examined by sequencing analysis. HAV-positive samples were analyzed as described by Taffon et al. [27] using a nested PCR with a broad range of primers targeting the VP1/2A junction of the HAV genome (267 bp amplicon).

AdV-positive samples were amplified with a seminested PCR using primers described by Allard et al. [28] with the KAPA Hotstart Ready Mix PCR Kit (Kapa Biosystems Inc. Wilmington, MA, USA) and the following thermal profile: 95 °C for 15 min followed by 35 cycles of 94 °C for 30 s, 68 °C for 15 s and 72 °C for 15 s, and a final extension at 72 °C for 5 min for both PCR reactions.

## 3. Results

Over a 36 month period, 289 samples of bivalve mollusks were tested from 20 shellfish farms located in three areas of the Campania region: two located in the estuary of ASL NA 1 Center, 10 in the estuary of ASL NA 2 North, and 8 in the ASL NA 3 South Estuary (Figure 1; Table 1). All shellfish samples were examined for the presence of HAV, NoV (GI and GII strains), AiV-1, AdV, AsV, SaV, RV, and HEV nucleic acid sequences. The number of positive samples for each target is shown in Table 4.

Of all samples tested, 159 were positive for virus contamination, with 62% of these positive samples containing at least two different virus RNAs (Table 5).

Overall, HAV was detected in 26 (8.9%) of the 289 examined samples. As shown in Table 4, the annual HAV prevalence was 4.7% (7/146), 5.1% (4/77), and 22.7% (15/66) in the years 2015, 2016, and 2017, respectively. HAV-positive shellfish showed a seasonal pattern, with samples only testing positive between December and April (Table 6).

Sequence analysis only gave good results for two of the HAV positive samples analyzed and they were classified as genotype IA, showing a 99% nt identity with strains observed in different Italian Regions in 2016 to 2017 (Accession numbers KY886891, KY292290, KY782330) during the European outbreak of hepatitis A among men who have sex with men (MSM) [29,30,31]. NoVGI strains were detected in 31 samples (10.8%), while NoVGII was detected in 114 samples (39.7%). For the other enteric viruses, RV was found in 26 (9.0%), AsV in 60 (20.8%), AdV in 16 (5.6%), SaV in 54 (18.8%), and AiV in 16 (5.6%). The AdV sequence obtained from the only sequenceable sample showed a 100% nucleotide identity with human AdV 41 strains already deposited in GenBank. HEV was never detected in any samples analyzed in this study.

Positive samples were found in both class A and B areas as shown in Table 7. Specifically, 39% of the samples collected in class A areas and 58% of class B samples were positive for at least one enteric virus.

Quantitative analysis showed different results for the viruses investigated. Specifically, HAV, NoVGI, and AdV displayed quantitative values/g of digestive tissue under the limit of quantification for almost of the samples. Only one sample exhibited an HAV RNA quantity of 4.2 × 10^2^ copies/g.

Notably, NoVGII showed quite a high average concentration value (1.1 × 10^6^ copies/g), with three samples reaching a concentration of more than 10^7^ copies/g. RV and AsV positive samples revealed a similar viral RNA average concentration, 1.9 × 10^3^ and 1.4 × 10^3^ copies/g, respectively. SaV and AiV yielded an average quantity of 5.3 × 10^2^ and 3.4 × 102 copies/g, respectively; five samples reached 10^3^ copies/g for SaV, and one sample 10^4^ copies/g, while AiV was found at the same concentration in all samples analyzed.

## 4. Discussion

A general trend of higher viral pathogen prevalence was noted in samples obtained in 2017. It is noteworthy that more intense rainfall occurred in southern Italy during the February to May 2017 period as compared to previous years. High rainfall and flooding are known to increase the risk of virus contamination of shellfish. Overall, the prevalence of HAV-positive shellfish was 8.9% for the three-year study. The HAV prevalence was not always homogeneous over time (Table 4). In 2017, the number of HAV-positive samples increased compared to the previous two years, from 4.7% in 2015 to 22.7%. However, there was no clear increase in human cases of hepatitis A associated with the consumption of raw mollusks in 2017. However, in 2014 to 2015, there were several reported shellfish-associated hepatitis A outbreaks in Italy, resulting from the consumption of raw and undercooked mollusks [32]. Regarding the other enteric viruses analyzed, the seasonal trend showed that winter had the higher level of virus positivity in shellfish, with the exception of rotavirus, which showed positivity all year (Table 6).

In this study, three HAV-positive samples were identified in class B areas between January and April 2017 and also one HAV-positive in a class A mussel area in which samples were collected in February 2015 from a Campania production area. This finding highlights the potentially serious hepatitis risk to consumers because class A shellfish are considered the most sanitary, which are frequently consumed raw. These data underline the necessity of consumer education programs to encourage people to avoid the consumption of raw shellfish.

Overall, we believe these data may be indicative of the continual release of HAV in the seawater ecosystem as it is shed by the population. HAV-positive samples were more often identified in the winter and early spring periods (Table 6). This pattern has been observed in previous studies in Italy and other EU countries [2,33]. For HAV samples, in general, low C_t_ signals between 38 > C_t_ < 40 precluded determining the sequence and HAV genotype of most positive samples. However, two HAV sequences were genotype IA, which is not the genotype traditionally found within shellfish in southern Italy. Whether this finding is a reflection of an increased genotype IA strain presence within the male homosexual population or due to another cause remains to be determined.

As for the viruses associated with gastroenteritis, NoV is considered a virus of primary concern for shellfish consumers, while other viruses (AiV, SaV, AsV, AdV, and RV) are either sporadically-associated with food-borne illness or are currently viewed as indicators of the sanitary quality of shellfish and shellfish growing waters [10]. In this study, NoVGII was present in samples at a substantially higher percentage (39.7%) than NoVGI (10.8%). Fusco et al. (2017) [2] observed this frequency pattern previously. The most probable explanation for this is that the NoVGII strains have a high prevalence in the community, where the virus is frequently transmitted person-to-person [34] and that this greater presence translates to a greater fecal viral load of GII strains as opposed to GI strains, which are less prevalent [35,36]. In addition, while for other viruses, there was a higher prevalence in 2017, this was not the case for NoVGII, which had a higher prevalence in 2016. Why NoVGII was more prevalent within shellfish in 2016 when there was less rainfall remains to be determined.

In the three-year period of the current study, 16 AiV-positive samples were identified, with a prevalence of 5.6%, and specifically, the prevalence trend over the three-year period ranged from a minimum of 2.6% in 2016 to a maximum of 7.8% in 2017 (Table 4). These figures contrast with a 12.0% frequency previously observed for mussels collected in the period of 2014 to 2015 for class A and B areas of the Tyrrhenian Sea [2], but are similar to the 6.0% prevalence observed in Galicia, Spain [37]. For this study, it appears that AiV contamination was influenced by weather conditions, since 2016 was the driest year of the three-year study and 2017 had the most rainfall. SaV was present in 54 (18.8%) mollusk samples. 

During the 36-month sampling period, virus detection in the samples progressively increased from 11.0% in 2015 to 26.5% in 2017 (Table 4). The high levels of SaV found within shellfish in this study are not unprecedented in Europe. In a study conducted on mussels collected in estuaries near the city of Coruña in Galicia, SaV was found in 30 out of 80 samples examined (37.5%) [22].

During the three-year period, the percentage of RV-positive samples was approximately 12% in 2015 and 2017, while in 2016, there were no positive samples. These results are similar to a study conducted in 2014 to 2015 from the same area, which noted an RV prevalence of 12.9% [2]. In the current study, AsV was present by PCR in 60 (20.8%) samples. This value is in general agreement with the findings of a previous Italian study [2] but are higher than frequencies reported by Vilariño et al. (2009) [38] in mollusks collected in Rio de Vigo, Spain in 2009. In the current study, 16 (5.6%) samples were positive for AdV. Sequence data from one sample that was positive by real-time PCR for AdV identified it as serotype 41, and in fact, human adenovirus species F serotype 41 is an important cause of acute gastroenteritis [23]. No samples were positive for HEV, although the presence of HEV in the wild ecosystem of the Campania region has been reported [39].

## 5. Conclusions

In conclusion, results from this study confirmed the circulation of enteric viruses pathogenic for humans in mollusks. It is very important to underline that virus positivity was also found in samples harvested in class A production areas. Nevertheless, the amount of positivity from class A production sites was lower than class B sites, with 20 positive samples out of a total of 46 samples from the class A sites analyzed and 136 positive samples out of 237 samples from class B (Table 7) These results underline an important question about the food safety of shellfish belonging to class A since according to Reg. CE 854/2004 [17], these mollusks could be directly consumed and do not need to be depurated, whereas class B shellfish require a process of depuration prior to sale, which is known to be, in any case, ineffective against viruses [33].

However, apparently, there have not been any major outbreaks or increases in the incidence of human illness caused by these viruses in recent years. It is possible that in the three-year period of this study, the surveillance activity carried out by veterinary services, together with information campaigns that have increased consumer awareness regarding the importance of cooking seafood, has achieved the goal of limiting human illnesses and as a consequence, facilitated a reduction of alerts from viral enteric health emergency groups regarding the transmission of viruses through consumption of contaminated mollusks.

## Figures and Tables

**Figure 1 ijerph-16-02588-f001:**
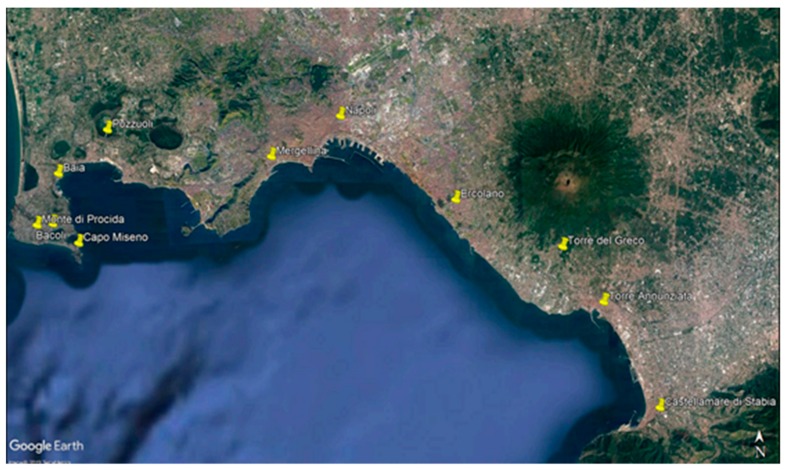
Shellfish sampling area (Campania region: Southeast Italy). The sampling areas are indicated by yellow dots.

**Table 1 ijerph-16-02588-t001:** Sample collection sites.

Collection Area	Number of Samples Collected	Percentage of Samples Collected (%)
ASL Na 1 Centre	27	9.4
ASL Na 2 North	201	70.0
ASL Na 3 South	56	19.9
Collection area not indicated	3	1.0

**Table 2 ijerph-16-02588-t002:** Classification of shellfish samples (as defined from Reg CE 854/2004).

Classification	Number of Samples Analyzed	Percentage of Samples Analyzed (%)
Class A ^1^	46	16.0
Class B ^2^	237	82.1
Class C ^3^	1	0.3
Natural banks ^4^	5	1.7

^1^ Class A: minimum of 10 samples required per year: 80% of sample results must be less than or equal to 230 *E. coli*/100 g and no results may exceed 700 *E. coli*/100 g. ^2^ Class B: minimum of eight samples required per year: 90% of sample results must be less than or equal to 4600 *E. coli*/100 g and no results may exceed 46,000 *E. coli*/100 g. ^3^ Class C: minimum of eight samples required: all sample results must be less than 46,000 *E. coli*/100 g. ^4^ Natural banks: Production area not yet classified.

**Table 3 ijerph-16-02588-t003:** Primers and probes for qPCR and RT-qPCR.

Target	Primers/Probe Name	Sequence	Reference
HAV Primer forward	HAV 68	5’TCACCGCCGTTTGCCTAG3’	
Primer reverse	HAV 240	5’GGAGAGCCCTGGAAGAAAG3’	[20]
Probe	HAV 150 FAM	5’CCTGAACCTGCAGGAATTAA-3’-MGB/NFQ	
NOROVIRUS GI			
Primer forward	NGI QNIF4	5’-CGCTGGATGCGNTTCCAT-3’	
Primer reverse	NGI NV1LCR	5’CCTTAGACGCCATCATCATTTAC3’	[20]
Probe	NVGG1P-FAM	5’TGGACAGGAGAYCGCRATCT3’TAMRA	
NOROVIRUS GII			
Primer forward	NGII QNIF2	5’ATGTTCAGRTGGATGAGRTTCTCWGA-3’	
Primer reverse	NGII COG2R	5’-TCGACGCCATCTTCATTCACA-3’	[20]
Probe	NGII QNIF FAM	5’-AGCACGTGGGAGGGCGATCG-3’TAMRA	
HEV			
Primer forward	JVHEV1	5’-GGTGGTTTCTGGGGTGAC-3’	
Primer reverse	JVHEV2	5’-AGGGGTTGGTTGGATGAA-3’	[23]
Probe	JVHEV P-FAM	5’-TGATTCTCAGCCCTTCGC-3’TAMRA	
AICHIVIRUS			
Primer forward	AiV-AB-F	5’-GTCTCCACHGACACYAAYTGGAC-3’	
Primer reverse	AiV-AB-R	5’-GTTGTACATRGCAGCCCAGG-3’	[21]
Probe	AiV-AB-TP FAM	5’-TTYTCCTTYGTGCGTGC- 3’NFQ (MGB)	
ROTAVIRUS			
Primer forward	NSP3F	5’ACCATCTWCACRTRACCCTCTATGAG-3’	
Primer reverse	NSP3R	5’-GGTCACATAACGCCCCTATAGC-3’	[22]
Probe	NSP3P-FAM	5’-AGTTAAAAGCTAACACTGTCAAA3’(MGB)	
SAPOVIRUS			
Primer forward	SAV124F	5’-GAYCASGCTCTCGCYACCTAC-3’	
Primer reverse	SAV1245R	5’-CCCTCCATYTCAAACACTA-3’	[24]
Probe	SAV124TPFAM	5’- CCCCTATRAACCA-3’NFQ (MGB)	
ADENOVIRUS			
Primer forward	AdV1	5’-CWTACATGCACATCKCSGG-3’	
Primer reverse	AdV2	5’-CRCGGGCRAAYTGCACCAG-3’	[25]
Probe	Advs-FAM	5’CCGGGCTCAGGTACTCCGAGGCGTCCT-3’	
ASTROVIRUS			
Primer forward	AsV1	5’-CCGAGTAGGATCGAGGGT-3’	
Primer reverse	AsV2	5’-GCTTCTGATTAAATCAATTTTAA-3’	[26]
Probe	AsV FAM	5’CTTTTCTGTCTCTGTTTAGATTATTTTAATCACC-3’ TAMRA	

HAV = hepatitis A virus; NGI = norovirus GI; NGII = norovirus GII; HEV = hepatitis E virus; AiV = aichivirus; SaV = sapovirus; AdV = adenovirus; AsV = astrovirus; FAM/TAMRA = fluorocrome; MGB = minor groove binder probe.

**Table 4 ijerph-16-02588-t004:** Positive results in the samples for various targets during the three-year testing period.

Viruses Detected	2015 Number of Positive Samples (%)	2016 Number of Positive Samples (%)	2017 Number of Positive Samples (%)	Total Positivity over Three Years (%)
HAV	7 (4.7)	4 (5.1)	15 (22.7)	26 (8.9)
NoVGI	6 (4.1)	7 (9.0)	18 (28.1)	31 (10.8)
NoVGII	31 (21.2)	53 (68.8)	30 (46.8)	114 (39.7)
RV	18 (12.3)	0 (0)	8 (12.5)	26 (9.0)
SaV	16 (10.9)	20 (25.9)	17 (26.5)	54 (18.8)
AsV	30 (20.5)	11 (14.2)	19 (29.0)	60 (20.8)
AiV	9 (6.1)	2 (2.6)	5 (7.8)	16 (5.5)
AdV	4 (2.7)	6 (7.7)	6 (9.3)	16 (5.5)
HEV	0 (0)	0 (0)	0 (0)	0 (0)
Total samples analyzed	146	77	66	289

RV = rotavirus.

**Table 5 ijerph-16-02588-t005:** Sample positivity.

Positivity of Samples	Number of Positive Samples	Percentage of Samples out of the Total Positive (%)
Positivity for one target	60	37.7
Positivity for at least two targets	99	62.2
Total positivity	159	55.0

**Table 6 ijerph-16-02588-t006:** Seasonal trend for virus positive samples from 2015 to 2017.

Month	Virus Positivity Number of Positive Samples/Number of Samples Analyzed (%)	HAV %	NoVGI %	NoVGII %	RV %	SaV %	AsV %	AiV %	AdV %
Jan.	23/28 (82.1)	25	12.9	20.1	0	12.9	10	25	3.7
Feb.	22/23 (95.6)	3.8	29.0	17.5	7.6	1.8	26.6	18.7	18.7
Mar.	18/26 (69.2)	11.5	9.6	8.7	0	12.9	20	12.5	6.2
Apr.	33/44 (75)	23.0	19.3	21.9	15.3	11.1	30	18.7	6.2
May	14/24 (58.3)	11.5	9.6	8.7	15.3	5.5	10	6.2	0
Jun.	3/4 (75)	0	0	1	0	0	1.6	0	0
Jul.	8/33 (24.2)	0	6.4	1.7	7.6	5.5	1.6	0	0
Aug.	0/13 (0)	0	0	0	0	0	0	0	0
Sept.	9/27 (33.3)	0	6.4	0	26.9	5.5	0	0	6.2
Oct.	0/18 (0)	0	0	0	0	0	0	0	0
Nov.	8/18 (44.4)	0	9.6	2.6	11.5	3.7	0	0	0
Dec.	22/28 (78.5)	15.3	64.5	17.5	15.3	11.1	5	25	25

**Table 7 ijerph-16-02588-t007:** Virus positivity among different sampling class areas.

Classification	Number of Samples Analyzed	Number of Positive Samples	Virus Identified
Class A	46	20	HAV, NoVGI, NoVGII, RV, AsV, SaV, Adv.
Class B	237	136	HAV, NoVGI, NoVGII, RV, AsV, SaV, Adv, AiV
Class C	1	1	HAV, NoVGI, NoVGII, RV, AsV, SaV, Adv, AiV
Natural banks	5	2	HAV, NoVGI, NoVGII, RV, AsV, SaV, Adv, AiV

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
