# Peer review of "Detection of Hepatitis A Virus and Other Enteric Viruses in Shellfish Collected in the Gulf of Naples, Italy"

_ijerph, 2019, doi:10.3390/ijerph16142588_

Round 1

Reviewer 1 Report

The introduction of the manuscript is well written, informative and puts forward a strong case regarding the environmental risk of acquiring food borne infections and the need for a common strategy to mitigate this risk. However, the experimental section is much beneath that.

One major weakness of the paper is that there are not actual viral loads reported, only positive/ negative which is unreasonable given a quantitative (RT-)PCR method was used, thus these data exist. Table 6 shows some data (only max and min loads), suggesting that less than 1 virus particle per gram of sample as being positive. This is puzzling to say the least. If the total amount of sample analyzed was 2 g, perhaps 10% of that used for quantification by qPCR thus authors claim that 0.01 RNA copies detected represent a positive result.

The methods described have been widely used by others, yet it is unclear whether these have been used according to the standards in the field, starting with sample size, need for internal amplification controls, etc. This puts in question the main conclusions of the manuscript.

Line 110 – “viral RNA was extracted from approximately 2 g of tissue” – is this enough to be considered a representative sample? The digestive gland of one animal (mussel) is then sufficient?

Were there internal amplification controls used? The sample was diluted 1/10 “to check for assay inhibition “ but how was this actually checked?  

Line 136 – “Quantification was carried out as previously described by Fusco et al. 2017.” Please explain briefly how this was done

Line 137-143 – please indicate if these were RNA or DNA standards and how they were used in the current work

Line 144-145 – are there ISO recommendations to support this decision? What was the Ct value of the standard with the lowest amount? If one of three is enough to consider positive, then why were triplicates used in the first place?

Line 146 – why was sequencing not attempted for all viruses?

Line 169-170 – this sentence is not supported by the data n table 6. Authors should show values of viral load per positive sample.

Author Response

Dear Reviewer,

We appreciate your very useful suggestions, and we hope we have satisfied them with our corrections (you can find them in the manuscript highlighted in gray)

Regarding the quantitavive analysis, we better explained the method that we used (See lines 133-136) in the Materials and Methods section. We decided to delete Table 6 (quantitative analysis of viruses detected) and  more clearly explained the results obtained in order to make the interpretation of data easier and more fluid for the readers. (See lines: 189-197)

Regarding the nucleic acid extraction, we explained this better on lines: 116-123; we hope that now we better clarified that 2 grams of hepatopacreas were in a pool obtained using digestve tissue of approximately twenty mussels  randomly selected from each sample. In addition, each RT-PCR assay for individual viral targets was performed in triplicate because each sample tested was a result of a pool of hepatopancreas so we can not consider the sample homogeneous.  Based on pervious research (Fusco et al 2017), we analyzed samples in triplicate in order to reduce the possibility of having false negatives.

As regards to the internal control; nucleic acids were extracted from 400 µl of viral extract using a mengovirus clone (vMC0) as the nucleic acid extraction efficiency control. UNI CEN ISO/TS 15216-2:2013

Comment regarding line 137-143:  We used quantified standard plasmids; please check  detailed correction highlighted in gray  on lines 137-143

Comment regarding line 146: Sequencing analysis was performed only for HAV and AdV beacuse only viruses not already characterized in previous works (Fusco et al. 2017) were further investigated.

Comment regarding lines 169-170: Table 4 shows the annual HAV prevalence in years 2015, 2016, and 2017, respectively, while Table 6 shows the seasonal trend for virus positivity.

We hope that we made the proper corrections and that our reply to the comments are satisfiactory.

Reviewer 2 Report

The study was to investigate the enteric viruses prevalence in shellfish from Gulf of Naples, Italy. In my opinion, the study was properly designed. The methods and results were clearly described in the manuscript with scientific discussion.

I would suggest authors perform data analysis and statistical comparison of the prevalence data to see if they are significantly increased/changed.

Author Response

Dear Reviewer,

The authors thank you very much for your comments, which have helped us to improve our manuscript.

 We have done our best to make the suggested English corrections.

Reviewer 3 Report

The article concern the presence of various enteric viruses in shellfish collected during three years in the gulf of Naples, underling the deviation of the results obtained from this study with the official analyses that use other parameters to monitor faecal contamination of the environments in which molluscs are farmed or harvested. The topic is of primary importance as could lead the legislator to reconsider which parameter is more suitable to classify the contamination of water and to establish new food safety parameters.

Beside all this aspects the manuscript need to an improvement before to be considered suitable for the publication on International Journal of Environmental Research and Public Health.

General comment.

The first effort that authors have to do is to increase the quality of English language that is in some case poor, generating a difficulty in understanding some sentences.

In several parts of the manuscript there is a digression from the central topic of the study through a digression concerning the diffusion of hepatitis A among homosexuals that is, in my opinion, not appropriate and it did not add useful information to the discussion of the presence of enteric virus in shellfish.

The presence of virus collected in class A and B areas must be better discussed with a specific table dedicated to explain better this topic as it has a considerable importance. Still now this aspect is only mentioned in the discussion without a specific comment.

Is it possible to introduce the data coming from E. coli determination to compare this results with your findings?

Conclusions must be more specific concerning the two different approach of enteric viruses and enteric bacteria for the classification of faecal contamination of waters.  

Specific comment

Line 6-7, some letters that identify affiliation are not in superscript

Line 22-23 the sentence must rewritten as it appears confused

Line 28 replace “not” with “never”

Line 63-66 the sentence must rewritten as it appears confused

Line 83 replace “bodies of water” with “waters”

Line 84, delete “the”

Line 90 replace with “Present study focuses with the detection of …”

Line 103 describe water characteristic of natural banks

Figure 1 -Eliminate marker (Vesuvio) on the maps as they could be confused with the sampling points. The marker Giugliano di Campagna indicate a sampling point? It is strange because it is not by the sea.

Table 7 insert the number of samples analysed each month How many samples were analysed in august? And October when there was none viruses positivity? The column viruses positivity was expressed as percentage?

Author Response

Dear Reviewer,

We thank you very much for your comments that have helped us to improve our manuscript.  We hope that we have adequately addressed them.

English corrections were made by the two native English speaking authors, and we hope that the corrections are now adequate.

Regarding the homosexual issue, we respectfully disagree with your opinion since we think this is important because we find an increased presence of the HAV strain (IA) circulating in Southwest Italy that was peviously uncommon within shellfish.  In the 2016-2017 period in Italy, 618 cases of acute hepatitis A were reported.  Recent HAV outbreaks in Central Italy (Lazio region; that is very  close to Naples) involved a high proportion of men (69.5%) suggesting homosexual transmission (Puoti et al.,2018). Furthermore, as of September 2018, the number of outbreak-confirmed cases reported in 22 EU/EEA countries since June 2016 was 475, all of which appear to be closely-related to genotype IA strains (ECDC, 2017; ECDC, 2018).

The degree to which this outbreak contributed to the high prevalence of HAV detected within shellfish in 2017 is unknown. Also, why there were no shellfish outbreaks reported during this period may be thanks to the surveillance activity carried out by veterinary services together with information campaigns that encourage consumers to avoid the consumption of raw shellfish.

Regarding the virus positivity among different class areas we have a specific table as you can see in the manuscript (Table 7). We hope that we safisfied your request.

I am so sorry but it is not possible to introduce the data on E. coli determination because this analysis was not performed on the shellfish. We explained the criteria of classification of water a (see Table 2: classification of shellfish samples) according to Reg. CE 854/2004 that gives an indication of sanitary quality of different Classes of shellfish producing waters. Nevertheless it is well understood that E.coli is not considered a good specific indicator for virus contamination since viruses can persist   within shellfish longer than E. coli. (L. Miossec; F. Le Guayader et al 2001: Validity of E. coli enterovirus and F-specific Rna Bacteriofages as indicators of viral shellfish contamination – Journal of shellfish research vol.20 n 3 1223-2227, 2001 / Samul R. Ferrah: Bacterial Indicators of Viruses- Part of the Food Microbiology of Food Safety book “Viruses in Food” Chapter 7 pages 189-204).

We addressed all of your specific comments as shown throghout the manuscript (highlighted in gray) and we modified Figure 1 and also Table 7 that now is changed to Table.6: Seasonal trend for virus positive samples from 2015 to 2017.

Round 2

Reviewer 1 Report

This version of the manuscript shows improvements on the methods section, with the extra added details. Of note is that authors quote themselves to justify how data analysis was done (line 142-3). Still one relevant question is why is the Ct value of 40 what determines positivity if standards with a known concentration were used. What is the average Ct value of 1 copy in each qPCR detection? A Ct value below 1 copy per reaction should be considered below the limit of quantification and not “positive”. Given that “Positivity” is the main criteria for analysis of the results of this manuscript and affects its main conclusions, this should be corrected.

Author Response

Dear Reviewer,

I am sorry about the misunderstanding but there was a typing error in lines 144-145, because  samples with a Ct value <40 (Ct<40) and not Ct≤40 were considered positive  to the presence of enteric virus while samples with a Ct value ≥40 and not >40 were considered negative. We made further analysis on the criteria of positivity and all samples considered positive had a Ct value <40.

We decided on this limit of detection because in our laboratory (Istituto Zooprofilattico Sperimentale del Mezzogiorno, Portici, Naples- Italy) we made different validation protocols for each target and we noticed that with a Ct value < 40 we were able to see an amplification curve.  We also decided to set a limit of detection at Ct <40 because we want to be more conservative and restrictive.

We thank you so much again for your comments that have helped us to improve our manuscript.  We hope that we have adequately addressed them.

Reviewer 3 Report

I thank the authors for having incorporated my indications and I appreciate the improvements brought to their article. I'm still not totally convinced about the relevance of the outbreaks of casis of Hepatits A among male homosexuals since there is not a correlation among this outbreaks and the presence of enteric viruses in shellfish.

Author Response

Dear reviewer,

we are so happy that you appreciate the review that me made as you suggest us.

I would like to thank you again for your effort to improve ours manuscript.